# New Circulating Variants of SARS-CoV-2 in Asturias During the Period (2022–2024)

**DOI:** 10.3390/v17121531

**Published:** 2025-11-22

**Authors:** José María González-Alba, Zulema Pérez Martínez, Susana Rojo-Alba, Cristina Ochoa Varela, Juan Gómez de Oña, Mercedes Rodríguez Pérez, Santiago Melón García, Marta Elena Álvarez-Argüelles

**Affiliations:** 1Unit of Virology, Microbiology Department, Hospital Universitario Central de Asturias, 33011 Oviedo, Spain; jmgonzalba@yahoo.es (J.M.G.-A.); zperezmrtnz@gmail.com (Z.P.M.); ssnrj4@gmail.com (S.R.-A.); crisochoavarela@gmail.com (C.O.V.); juan.gomezde@sespa.es (J.G.d.O.); mercedes.rodriguezp@sespa.es (M.R.P.); santiago.melon@sespa.es (S.M.G.); 2Instituto de Investigación Sanitario del Principado de Asturias (ISPA), 33011 Oviedo, Spain; 3Genetic Department, Hospital Universitario Central de Asturias, 33011 Oviedo, Spain

**Keywords:** SARS-CoV-2, new variants, surveillance, phylogenetic analysis

## Abstract

The ability of a virus to adapt is key to its survival, and this is achieved through mutation, which allows the virus to change and adapt to new environments. To capture the full extent of SARS-CoV-2 diversity in Asturias, samples obtained from nasopharyngeal swabs were characterised using whole-genome sequencing. Between 2020 and July 2024, a total of 4001 sequences were analysed and 5302 mutations were identified. An increase in the positivity rate was observed between 2022 and 2024 in children under 1 year of age. During this period, 55 new circulating variants belonging to 41 pangolin lineages were detected: 24 originated throughout the world and 31 in Asturias (10 detected only in the region, 8 in the rest of Spain, and 13 around the world). A total of 31 new non-synonymous mutations were fixed in the viral population 250 ± 46 (93–620) days after their appearance. During seasonal SARS-CoV-2 circulation, surveillance systems developed during the pandemic continue to detect new indigenous and imported variants without indicating an increase in severity.

## 1. Introduction

Since the start of the pandemic, the rate of mutation of SARS-CoV-2 has been decreasing [1]. However, the number of SARS-CoV-2 infections continues to rise, and outbreaks still occur despite the majority of the population being vaccinated. It is fortunate that the current fatality rate due to COVID-19 is lower than it was in the initial phase of the pandemic [1]. This is because the causative agent of COVID-19, SARS-CoV-2, is constantly evolving as it spreads from person to person, with new sub-lineages emerging all the time. Consequently, the genome of the virus should be analysed in order to identify the viral strains in circulation and to investigate their dissemination both geographically and worldwide. Furthermore, the comprehensive sequencing of the entire viral genome associated with infection is a valuable tool with which to elucidate outbreak dynamics [2,3]. Our group started sequencing the whole genome of the SARS-CoV-2 samples to identify the genotypes of the virus circulating in our area and to analyse genomic diversity, the types of mutations, and the emergence of new variants of SARS-CoV-2 by June 2022 [4,5,6]. In June 2022, the Spanish Ministry of Health proposed infection control and surveillance measures that reduced pressure on cases of mild or asymptomatic illness and their contacts [5]. Consequently, all respiratory viral illnesses, including those caused by SARS-CoV-2, must now be considered as posing the same risk. This study monitored and tracked the SARS-CoV-2 epidemic in Asturias during the post-COVID-19 period (July 2022 to July 2024).

## 2. Materials and Methods

### 2.1. Sample Collection

From July 2022 to July 2024, 2268 samples (1733 recorded from March 2020 to July 2022) obtained from nasopharyngeal swabs from individuals infected with SARS-CoV-2 were characterised by means of the whole-genome sequencing (WGS) method and were uploaded to GISAID. Data on age, sex, date, and pangolin lineage were collected (Appendix A).

### 2.2. WGS

Selected SARS-CoV-2 positive samples were sequenced using the Ion AmpliSeq SARS-CoV-2 research panel (Thermo Fisher Scientific, Waltham, MA, USA) following the instructions set out in the manufacturer’s user guide. Libraries were prepared on the Ion Chef system in accordance with the guidelines outlined in the user’s guide. Subsequently, the amplified samples were subjected to sequencing using Ion 540 chips with the Ion S5 system. This process was carried out in strict accordance with the manufacturer’s user guide.

The obtained sequences were uploaded to the GISAID database (https://www.gisaid.org/).

### 2.3. Classification/Characterisation

The SARS-CoV-2 genomes were aligned using MAFFT (https://mafft.cbrc.jp/alignment/software/ (accessed on 30 January 2022)) and then manually curated using MEGA 7 (https://www.megasoftware.net/ (accessed on 16 June 2022)). The nucleotide substitution model (GTR + I) was determined using the Akaike information criterion with jModelTest v2.1.10 (https://github.com/ddarriba/jmodeltest2 (accessed on 16 June 2022)). Phylogenetic trees were reconstructed by ML with FastTree (http://www.microbesonline.org/fasttree/ (accessed on 16 June 2022)) for large trees or IQ-TREE (http://www.iqtree.org/ (accessed on 22 February 2022)). Bootstrap values were estimated using the SH test and ultrafast bootstrap. The Wuhan-Hu-1 reference genome (MN908947.3) was used as an outgroup.

The coding regions (ORF1a, RdRp(RNA-dependent RNA polymerase), ORF1ab, S, ORF3a, E, M, ORF6, ORF7a, ORF7b, ORF8, N, and ORF10) were extracted individually from the alignments. Subsequently, the nucleotides in the coding regions were converted to their corresponding encoded amino acid residues (SeaView https://doua.prabi.fr/software/seaview (accessed on 22 February 2013)). The retrieval of SNPs was conducted within the aligned regions, in accordance with the reference genome. Non-synonymous mutations with a frequency of more than 5% (number of strains with a specific mutation/total number of strains) were considered the majority in the population and were used in subsequent analyses. A mutation is considered to have become fixed in the population when it reaches a frequency of 95% (number of strains with a specific mutation/total number of strains in a month) and remains until today.

A dated phylogeny was reconstructed using Bayesian inference via a Markov chain Monte Carlo (MCMC) framework in BEAST v1.10 (https://beast.community/ (accessed on 20 April 2022)). An uncorrelated relaxed clock model was employed to estimate the time to the most recent common ancestor (TMRCA). MCMC chains were run for 100 million steps (sampling every 10,000). Convergence was evaluated using Tracer v1.7.1 (https://beast.community/tracer (accessed on 19 November 2025)).The trees were summarised using TreeAnnotator v1.8.4 after the first 10% were discarded as burn-in and then visualised in FigTree v1.4.4 (http://tree.bio.ed.ac.uk/software/figtree/ (accessed on 15 November 2021)).

A phylogeographic analysis was performed in the BEAST programme. An asymmetric substitution model and an uncorrelated relaxed molecular clock were applied to the Bayesian stochastic search variable selection (BSSVS) method to identify the number of non-zero transition rates between states.

Possible new lineages were defined for study based on SNPs and monophyletic clades with more than five sequences (according to the proposed dynamic nomenclature for SARS-CoV-2 lineages at https://github.com/cov-lineages/pango-designation (accessed on 8 June 2022)). Lineage-specific mutations were obtained by analysing WGS genomes against the Wuhan-Hu-1 reference genome (MN908947.3). The pattern of mutations specific to the possible new lineages was sought in the sequences obtained from GISAID to analyse their global distribution.

The diversity (D = 1 − ∑f^2^) of the pangolin lineages was analysed, taking into account the frequencies (f) of all types.

## 3. Results

A total of 4001 sequences were analysed (randomly selected from 533,769 positive SARS-CoV-2 samples with ct < 27) in Asturias (northern Spain) between 2020 and July 2024. The positivity rate of SARS-CoV-2 by age in the pre-Omicron and Omicron periods is shown in Table 1.

### 3.1. Mutations

During these four years, 5302 amino acid changes were found in the 9769 amino acids that make up the complete genome. Of these, 157 (3% of mutations and 1.6% of the total genome) occurred in more than 5% of the viral population. In the same period, 158 (1.6% of the genome) mutations were found in Spain, and 116 (1.2%) were found worldwide. (Appendix A, Table 2).

The estimated relative mutation rates for the codon were 1.34 for the third position, 0.81 for the second position, and 0.85 for the first position.

The estimated mean rate was 9.20 × 10^−4^ (8.07–10.00) replacements per site, per year in the 2022–2024 period, compared to 7.92 × 10^−4^ (7.30–8.54) replacements per site, per year in the 2020–2022 period.

Between 2022 and 2024, 88 new non-synonymous mutations (0.9% of the genome) occurred in over 5% of the viral population, of which 31 are currently maintained and were fixed 250 ± 46 (93–620) days after their appearance. Between March 2020 and July 2022, 105 (1.1% of the genome) major mutations were identified, of which 51 were fixed after 344 ± 94 (0–1209) days (Appendix A, Table 3, Figure 1).

In contrast to Asturias, as of June 2024, 50 mutations have been established worldwide: 29 in gene S, 8 in ORF1a, 4 in the N gene, 2 in the M gene, 3 in ORF1ab, and 1 in RdRp, ORF3a, the E gene, and ORF6 (Appendix A).

The P323L mutation (RdRp gene) and the D614G mutation (S gene) appeared in over 5% of the viral population by day 0. The following mutations were delayed two months: M gene-A63T, M gene-Q19E, ORF1a gene_P3395H, ORF1ab gene_I643V, S gene_N679K, S gene_N764K, S gene_N969K, S gene_Q954H, and S gene_S373P.

Of the 82 mutations fixed, 41 occurred between February and June 2022, and 41 (10 of which appeared in the first period: N_Q229K, ORF1a_A2710T, ORF1a_T4175I, ORF6_D61L, S_A264D, S_E484A, S_G446S, S_L216F, and S_R21T, S_S939F) between January and June 2024 (Figure 2).

### 3.2. Lineages

From June 2022 to July 2024, 313 pangolin lineages were identified in Asturias, of which BA.5.1 (6.1%), DV.7.1 (5.8%),and BQ.1.1 (5.5%)accounted for over 5% of cases (see Appendix A, Figure 3). In this period, diversity reached a maximum of 0.980. During the first period (June 2020–2022), diversity was 0.929, and 111 lineages were identified.

From 2020 to 2024, 55 new circulating variants (NCVs) belonging to 41 pangolin lineages were detected (Appendix A). The distribution of the 91 mutations gained from these NCVs was 21 in NSP3 (ORF1a gene), 17 in S, 9 in ORF3a, 8 in NSP2, and 6 in NSP14 (Appendix A, Table 4).

Of these 55 NCVs, 24 were circulating throughout the world, with 2 (BF.5 + S_D80E + S_A701V + NSP1_G94C and JN.1.16.1 + NSP3_S1428L + NSP5_K90R + NSP6_L37F + S_F59S) mainly in Asturias, during 59 ± 16 (6–169) days; only variant XBB.1.5.8 + NSP14_V328F in March 2023 belonged to the same transmission clade in the region (Figure 4). The other 31 originated in Asturias: 10 of them were only detected in the region during 92 ± 20 (31–129) days, 8 in the rest of Spain during 107 ± 31 (43–154) days, and 13 around the world during 103 ± 20 (44–184) days (Table 5).

Figure 3C shows the variants circulating in the region, both imported and indigenous, at the time.

### 3.3. Variants of Interest

Of the 82 non-synonymous mutations fixed in Asturias by 2024, 17 were used to define a variant of interest (Table 6).

Figure 5 shows the number of fixed mutations found in each variant of interest in Asturias and the rest of the world (Appendix A).

## 4. Discussion

Viruses constantly change through mutation, and some changes allow the virus to spread more easily or make it resistant to treatments or vaccines. As the virus spreads, it may change and become harder to stop. The mutation rate may not be as high as one might think for the effects to be significant; regardless of the rate at which the environment fluctuates, the highest levels of adaptability occur at intermediate mutation rates [7]. In order to achieve effective surveillance, it is essential to obtain a sufficient amount of sequence data from a representative population. This data must be analysed in order to detect new variants and monitor trends in circulating variants. In 2022, only the impact of the disease on vulnerable people, hospitalisations, and deaths was monitored [8]. Consensus genomic sequences of SARS-CoV-2 variants represent the most frequently observed viral genomes in clinical samples from patients and are widely used to monitor the global spread of the virus [9,10]. However, they do not reflect the full diversity of viral genomes.

Currently, there are growing signs that adaptive evolution of SARS-CoV-2 has stalled, and purifying selection is the dominant evolutionary force acting on non-synonymous mutations in the Omicron lineage [11]. Because mutations at the third nucleotide of a codon often lead to no change in the amino acid sequence, the most frequent change occurs in the third position of the codon [12]. However, despite the fact that SARS-CoV-2 is an RNA virus with a high mutation rate, changes that are maintained only occur in 1% of the viral genome. As you might expect, these changes occur mostly in the S protein. However, mutations in the ORF1a gene, which encodes proteins involved in regulating the host response, are also notable [13,14].

In Asturias, we observed 157 mutations out of 5% of the viral population. This is a similar proportion to that in Spain and slightly higher if all the viral strains circulating in the world are taken into account, where 116 mutations were observed. This is because the number of viral strains is greater, making it more difficult to obtain that 5% viral population for a given mutation.

If we compare the latter period with the start of the pandemic, it is striking that the estimated mutation rate is higher in the post-pandemic era. However, of the non-synonymous mutations that occurred, 31 were fixed in the post-pandemic period, compared to 51 in the first period. This is because, once the initial changes are in place, subsequent changes are more difficult to establish, especially if the first mutations have evolutionary advantages.

In a previous study and as reflected by other authors [3,5,15], mutations can emerge, giving rise to new variants, as the number of infected people increases. This second study provides additional information: it appears that the mutations are fixed within 18 months of the onset period, as illustrated in Figure 2. It will be interesting to see if the pattern of mutation fixation is maintained in 2026 and the number of established mutations continues to decrease due to purifying selection.

High vaccination rates and the widespread use of face masks were considered important factors in preventing the emergence of variants and slowing the spread of the virus. However, the diversity of the virus remains, being greater than during the pandemic. In the post-pandemic era, 313 lineages were identified in Asturias, but only 4 represent more than 5% of the viral population. Furthermore, the gain of 91 mutations led to the classification of 55 new circulating variants according to the criteria for designation of a new Pango lineage (necessary but not sufficient).

As mentioned above, the most frequent mutations are in the S and ORF1a genes. Of the four structural proteins, only the S protein (17 mutations), which serves as the primary antigen targeted by the host immune response, accumulates changes in the new variants. Mutations in viral proteins, particularly the S protein, can significantly affect viral infectivity, virulence, and immunogenicity, so they require continuous monitoring. The RBD (and particularly the RBM) is the core part of the S protein that binds directly to ACE2 in host cells. Some mutations in the RBM (three mutations) can induce significant changes in SARS-CoV-2 phenotypes. Genetic mutations at the V445 site influence membrane fusion and entry into diverse target cells, mutations at the F456 site often result in resistance to class A antibodies, and mutations at the F486 site escape the epitopes of class B antibodies [16,17].P1263L shows a significant increase in fusion [18] compared to P1263Q.Most mutations that generate new variants occur in accessory proteins: ORF7a (3 mutations) can suppress the IFN-I response by inhibiting STAT2, and ORF8 (2 mutations) downregulates the presentation of viral antigens via the class I major histocompatibility complex [19]; the ability to suppress the IFN-I signalling pathway has been exhibited by ORF10 (1 mutation) through its interaction with the mitochondrial antiviral signalling protein [19,20]; ORF3a (9 mutations) modifies crucial cellular processes, such as apoptosis and autophagy [19,21]; inflammatory cytokines have been reported to be induced by both ORF3a and ORF7a through the activation of NF-κB signalling [21]; and Nsp2 (8 mutations) is involved in disruption of signalling in host cells [22]. More surprising are the abundant mutations in NSP3 (21 mutations), which is involved in promoting RNA replication, transcription, and cleavage of proteins involved in the host innate immunity [23,24].

The COVID-19 pandemic has shown a tendency to generate variants in specific geographic areas and cross-border transmission patterns [25,26,27,28]. The introduction and dissemination of the virus across different regions has been facilitated by tourism, transportation routes, and global migration patterns [29]. The increase in incidence may be accompanied by the appearance of new variants originating in the region. New variants circulating around the world continually appear in our region. This makes them the ones that are detected for the longest time, as they are more difficult to control, although none of them generate large outbreaks. In fact, the NCVs from Asturias are the majority. But over time, the absence of new local variants causes a rebound with imported variants. None of the predominant lineages in the epidemic has remained for long, and only two are actively transmitted in the region. This indicates a constant turnover of lineages, each of which predominates for a couple of months at most and without causing any serious problems.

Although most changes have little to no impact on the properties of the virus, this high mutation rate and the continuous appearance of NCVs require genomic surveillance, since some changes can lead to a more aggressive variant that can spread and predominate in the rest of the world. Therefore, lineages may retain mutations that became dominant in the global pandemic over time, which may have positive effects on the fitness of the virus and facilitate the emergence of new variants. The WHO has established a dedicated group to monitor the evolution of the virus. This group has been operational since June 2020 in tracking SARS-CoV-2 variants [30].

The Omicron variant has spread worldwide and is now the predominant variant. In our region, after the arrival of the Omicron variant and the transition to a strategy that reduces pressure on mild or asymptomatic cases and their contacts, there was a decrease in the sampling rate in the age group between 6 and 65 years; an increase in the positivity rate was observed in children under 1 year of age. Compared to previous variants of concern (VOCs), the Omicron variant is characterised by increased infectivity and immune evasion, as well as a substantially larger number of mutations [31]. Mutations originate from the previous variants within the RBD of Omicron; K417N and N501Y (Beta) are largely responsible for these monoclonal antibodies failing to neutralise the Omicron S [32]. Some of these mutations increased affinity for ACE2 (T478K -Delta-, N501Y), while others decreased ACE2 affinity (K417N) [31]. Other mutations fixed from previous variants of interest are as follows. The RdRp_P323L (Alpha)mutation is required for polymerase activity and is predicted to diminish the efficacy of antiviral drugs.The S_D614G (Alpha) mutation increases infectivity [33,34]. S_H655Y (Gamma) may have reduced neutralisation by monoclonal antibody therapies, convalescent sera, and post-vaccination sera [35];it is known to enhance viral growth and S protein cleavage by furin [36], and it also governs entry through endosomes, as suggested by the significant increase in viral infectivity [37]. N protein mutations, including P13L (Lambda), R203K, and G204R/K (Alpha),may increase the transmission of SARS-CoV-2, but they are also associated with reduced disease severity and lower mortality rates than individuals with the wild type [38,39,40]. P13L itself has been identified as the most important epidemiological driver of fitness in the N protein [41]. The S_E484K mutation, present in Beta, has reappeared; it escapes the neutralising effect of several monoclonal antibodies, convalescent plasma, and post-vaccine sera [42]. The ORF1a_T3255I (nsp4_T492I) mutation, present in Lambda, increases the virus’s replication capacity and infectivity and improves its ability to evade host immune responses [43].

The global SARS-CoV-2 pandemic is decreasing, but the populations affected by the virus continue to appear. Geographical epidemiological research clarifies how the virus is spread in the context of the COVID-19 pandemic, explaining how the pathogen propagates within and between populations, which helps to identify and contain chains of transmission [44,45]. Fortunately, all newly detected circulating variants appear to be extinct and do not exhibit signatures suggestive of adaptive or transmission-related changes, but further examination of the spatial patterns and transmission dynamics of COVID-19 at various scales, from local communities to global populations, is necessary to implement localised lockdown measures to contain the spread, the day a really dangerous variant is generated

## 5. Conclusions

Even as the global pandemic of SARS-CoV-2 recedes, the number of people affected by the virus continues to rise, making it crucial to maintain a strong focus on the issue. Controlling the impact of the virus on a global scale will require continuous efforts to understand and adapt to the ever-changing situation.

This study is going to help us trace the origins and sources of the different versions of the virus that are doing the rounds and compare the emerging variations within Asturias. The next lineage, which may be more transmissible, infectious, and able to evade vaccine-induced or natural host immunity, could be the result of low-frequency lineages and circulating VOCs.

Mutations that increase SARS-CoV-2 transmission and may be associated with less disease severity are those that are becoming established in the virus population.

In addition to direct intervention measures, sustained monitoring of SARS-CoV-2 plays a crucial role in controlling variants. Continuous surveillance facilitates the early identification of variants that undergo substantial changes in their adaptability, which allows rapid adjustments to preventive measures and will contribute to strengthening preparedness for the next pandemic. The discovery of new local variants that can be imported and exported emphasises the importance of local surveillance in informing public health responses relating to diagnostics, vaccination strategies, and health policies at local, regional, and global levels.

## Figures and Tables

**Figure 1 viruses-17-01531-f001:**
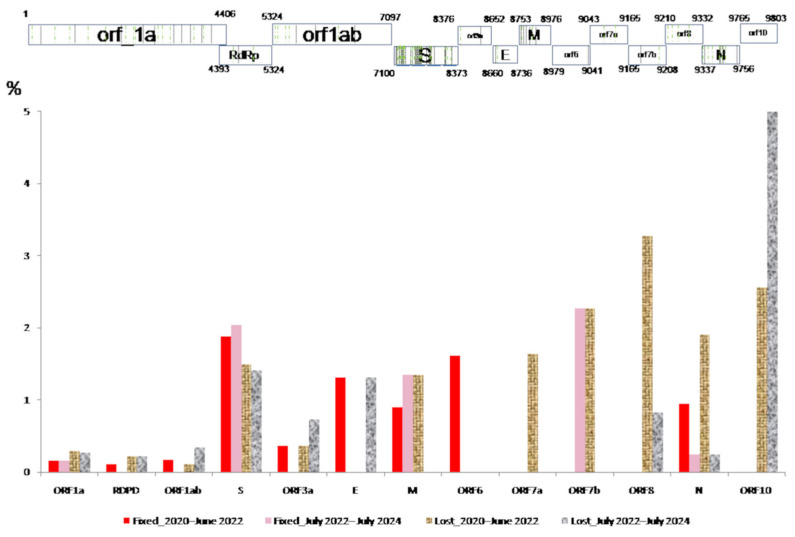
Rate of main mutations circulating in Asturias that have been fixed (

) and lost (

) in the population in each gene during COVID-19 (March 2020 to June 2022) and the post-COVID-19 period (July 2022 to July 2024).

**Figure 2 viruses-17-01531-f002:**
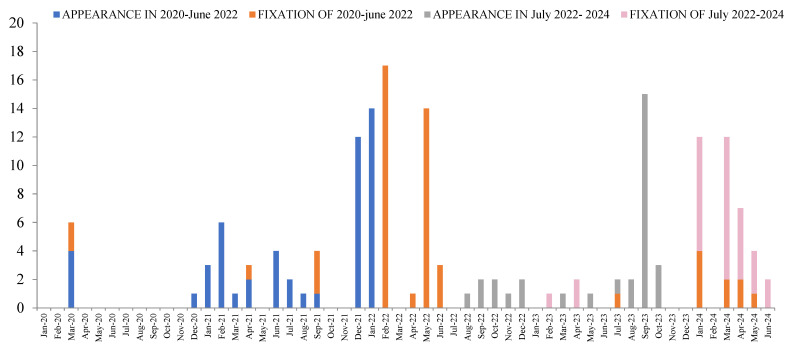
Number of major SARS-CoV-2 mutations (>5% of the viral population) that became fixed in Asturias during COVID-19 (March 2020 to June 2022) and the post-COVID-19 period (July 2022 to July 2024). The figure shows the dates of first detection and fixation.

**Figure 3 viruses-17-01531-f003:**
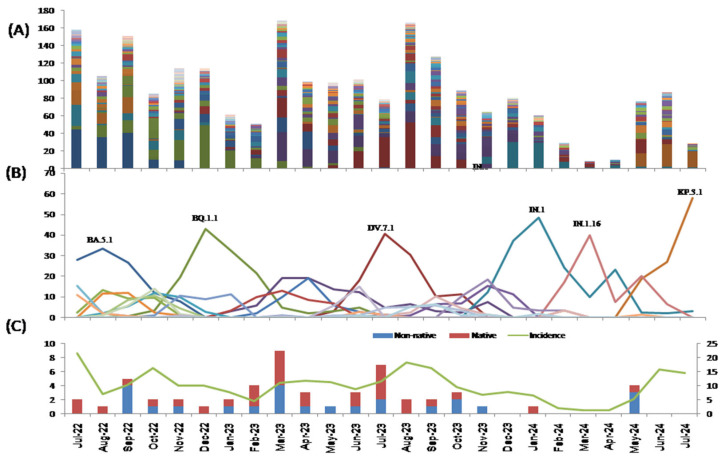
(**A**) Number of pangolin lineages (313 colour schemes not included, each colour represents a pangolin lineage) detected in Asturias in the post-COVID-19 period (July 2022 to July 2024). (**B**) Percentage of pangolin lineages mostly detected over time (>1%) in the post-COVID-19 period (July 2022 to July 2024);only the major lineages are indicated.(**C**) Number of new circulating variants that originated in Asturias or were imported and incidence of SARS-CoV-2 in the post-COVID-19 period (July 2022 to July 2024).

**Figure 4 viruses-17-01531-f004:**
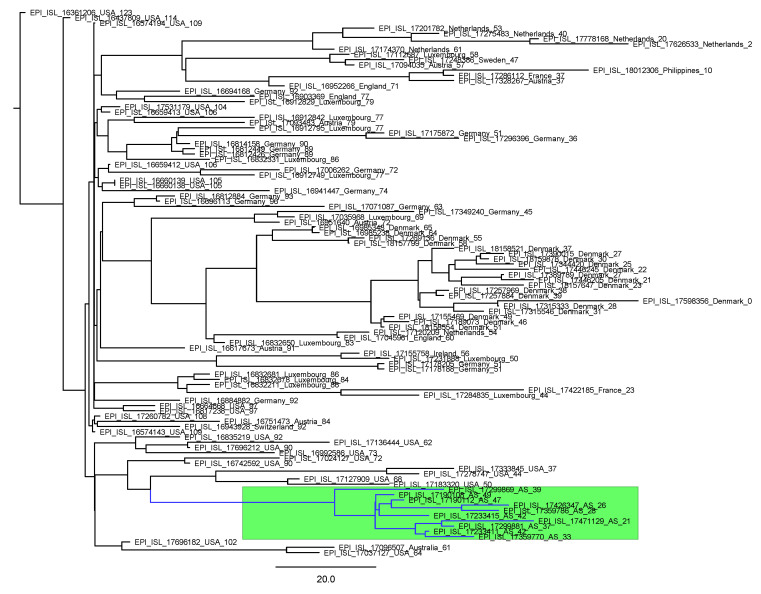
Dated phylogeny of the new circulating variant, XBB.1.5.8 + NSP14_V328F, which was imported and belongs to the same transmission clade in the region (green color in the figure). The days until the date of the most recent sequence were used as the sampling date.

**Figure 5 viruses-17-01531-f005:**
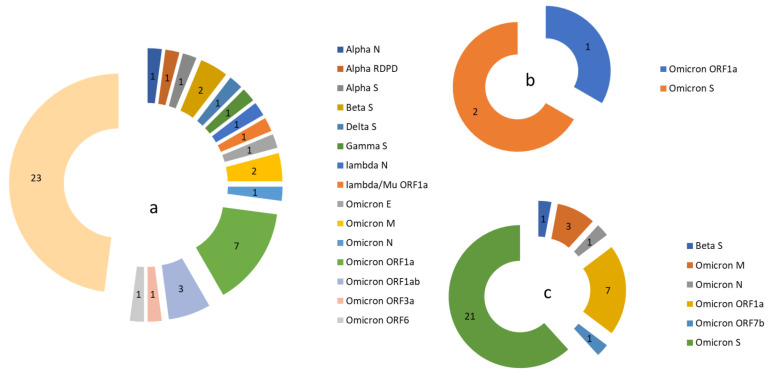
The number of major mutations per gene currently fixed and the variants of interest in which they were initially found. (**a**) In Asturias and in the world, (**b**) in the world but not in Asturias, (**c**) in Asturias but not in the world.

**Table 1 viruses-17-01531-t001:** Sequences sampled and positivity rate by age in the pre-Omicron period (until November 2021) and in the Omicron period (March 2022 to July 2024).

	Pre-Omicron Period		Omicron Period	
Age (Years)	Sampling	Positivity (%)	Sampling	Positivity (%)
<1	3481	2.7	4136	12
1–5	19,605	2.9	10,494	2.7
6–20	61,017	6.5	12,499	4.5
21–65	253,653	5.5	36,489	19.9
>65	100,920	4.8	31,475	16.1

**Table 2 viruses-17-01531-t002:** The number of mutations that occurred in >5% of the viral population sequenced in GISAID in each gene between 2020 and July 2024.

	ORF1a	RdRp	ORF1ab	S	ORF3a	E	M	ORF6	ORF7a	ORF7b	ORF8	N	ORF10	Total
GISAID	21	2	6	59	2	2	5	1	2	1	4	11		116
SPAIN	29	4	7	82	2	2	8	1	2	2	4	14	1	158
ASTURIAS	31	3	8	82	2	2	7	1	2	2	4	12	1	157

**Table 3 viruses-17-01531-t003:** Number of mutations fixed occurring in each gene in the Asturias strains during COVID-19 (March 2020 to June 2022) and the post-COVID-19 period (July 2022 to July 2024).Time is the average number of days it took to become fixed with its confidence interval.

Mutations		ORF1a	RdRp	ORF1ab	S	ORF3a	E	M	ORF6	ORF7a	ORF7b	ORF8	N	ORF10	Total
2020–2022	**Occurred**	21	3	5	47	2	1	5	1	2	1	4	12	1	105
	**Fixed**	8	1	3	29	1	1	2	1				5		51
	**Days or** **Mean** **± CI**	62/124(x4)/217/775,930	0	62 124 124	358 ± 132 (0–1209)	496	372	62 62	868				124/310 403/713,961		344 ± 94 (0–1209)
2022–2024	**Ocsurred**	20	3	6	47	2	1	3			1	1	2	2	88
	**Fixed**	6			21			3			1				31
	**Days or** **Mean** **± CI**	93/124(x2) 217(x3)			257 ± 53 (124–527)			248 278 620			217				250 ± 46 (93–620)

**Table 4 viruses-17-01531-t004:** Mutations gained by new circulating variants in Asturias.

ORF_1a									RdRp	ORF_1ab										
NSP1	NSP2	NSP3	NSP4	NSP5	NSP6	NSP8	NSP9	NSP10	NSP12	NSP13	NSP14	NSP15	NSP16	S	ORF3a	E	ORF7a	ORF8	N	ORF10
G94C	A357S	D112N	A128V	K90R	F235L	P10S	G38V	T115I	A4774V	V157L	A320T	D36G	K160R	A67V	D155Y	T9V	L5F	D119Y	P151S	I27T
P6S	E574D	D1764G	V13I	V35L	L37F		T21I		I4563M		I15T	P205S	P215L	A701V	D27H		Q94L	P38S		
V121A	G265S	E119K							M5021V		N71S			A845S	L140F		T28I			
	G285S	E387D							S4621N		P203L			D253G	M260K					
	L24F	I541V									Q22H			D80E	Q185H					
	N254S	M988L									V328F			F456L	Q57H					
	S591I	N1322S												F486V	S171L					
	T103I	N1680K												F59S	T270I					
		P153L												G252V	V273L					
		Q167R												L249F						
		R1297G												N354K						
		S126L												P1263Q						
		S1428L												Q675H						
		S454G												T547I						
		T1203I												T572I						
		T424N												V1264L						
		T720I												V445P						
		T970M																		
		V1385I																		
		V1673I																		
		Y1535H																		

**Table 5 viruses-17-01531-t005:** Asturian new circulating variants between 2020 and 2024.

NCV	N (%Total Identified)	First Detection Date	Last Detection Date	Time of Detection (Days)
Frst detected in the world				
BA.5.2.1 + NSP3_S454G	6 (43)	17/09/2022	22/10/2022	35
BA.5.2.6 + NSP3_M988L	7 (3)	22/09/2022	06/11/2022	45
BF.5 + S_D80E + S_A701V + NSP1_G94C	8 (62)	22/09/2022	08/11/2022	47
BA.5.1 + NSP12_M5021V	5 (7)	24/09/2022	16/11/2022	53
BF.7 + NSP14_P203L	13 (29)	19/10/2022	07/12/2022	49
BQ.1.1.18 + S_T547I	5 (28)	16/11/2022	11/01/2023	56
XBB.1.5 + NSP3_T1203I	12 (6)	24/01/2023	01/05/2023	97
FL.5 + NSP14_Q22H + NSP3_P153L	6 (2)	15/02/2023	24/05/2023	98
XBB.1.5.8 + NSP14_V328F	10 (10)	01/03/2023	29/03/2023	28
BQ.1.18 + NSP14_I15T	20 (17)	04/03/2023	29/06/2023	117
XBB.1.5 + NSP2_S591I + NSP3_N1322S	12 (1)	04/03/2023	20/08/2023	169
XBB.1.5 + NSP5_V35L	6 (1)	09/03/2023	07/07/2023	120
XBB.1.5.71 + NSP1_P6S + ORF3a_Q57H + NSP12_I4563M	5 (4)	28/04/2023	13/06/2023	46
XBB.1.5.71 + NSP1_P6S + NSP12_I4563M	6 (2)	12/05/2023	03/06/2023	22
EG.5.1 + ORF3a_D27H	5 (1)	20/06/2023	24/08/2023	65
XBB.1.16.11 + NSP2_N254S	6 (1)	20/07/2023	16/10/2023	88
DV.7.1 + NSP3_E119K	6 (18)	26/07/2023	08/09/2023	44
EG.5.1.3 + NSP3_I541V	5 (42)	05/09/2023	14/09/2023	9
JD.1.1 + ORF3a_M260K	5 (2)	13/10/2023	28/11/2023	46
JG.3 + ORF3a_S171L	7 (14)	27/10/2023	04/12/2023	38
JN.1.31 + S_T572I	6 (23)	29/11/2023	13/03/2024	105
JN.1.16 + S_A67V + S_L249F + S_V445P	6 (10)	07/05/2024	26/05/2024	19
JN.1.32 + S_F456L	7 (1)	12/05/2024	06/06/2024	25
JN.1.16.1 + NSP3_S1428L + NSP5_K90R + NSP6_L37F + S_F59S	5 (83)	13/05/2024	19/05/2024	6
First detected in Asturias				
BA.4.6 + NSP2_G265S + NSP3_D112N + ORF7a_Q94L	7 (100) **	10/06/2022	17/10/2022	129
CH.1.1.28 + NSP16_K160R	14 (100) **	22/12/2022	22/03/2023	90
XBB.1.5 + NSP3_T970M + NSP12_A4774V	6 (100) **	27/01/2023	13/04/2023	76
XBB.1.5.77 + S_G252V + NSP2_E574D + ORF7a_L5F + ORF7a_T28I	6 (100) **	14/02/2023	07/04/2023	52
XBB.1.5.1 + NSP15_D36G + NSP3_S126LL + NSP1_V121A + ORF8_P38S	7 (100) **	03/04/2023	04/05/2023	31
XBB.2.3 + E_T9V + NSP3_N1680K + NSP3_Y1535H + ORF10_I27T	5 (100) **	16/04/2023	11/08/2023	117
EG.1.4 + NSP12_S4621N + S_A845S	6 (100) **	09/05/2023	03/08/2023	86
DV.7.1 + NSP3_R1297G + ORF3a_Q185H	6 (100) **	21/05/2023	31/08/2023	102
JG.3 + NSP3_V1385I	8 (100) **	04/10/2023	22/01/2024	110
JN.1.16.2 + S_A67V + S_V445P + S_L249F	8 (100) **	24/01/2024	25/05/2024	122
BQ.1.1.15 + NSP3_T1203I	6 (55) *	26/07/2022	27/12/2022	154
BQ.1.1.66 + NSP3_E387D + NSP9_G38V + NSP10_T115II	10 (83) *	23/11/2022	15/03/2023	112
XBB.1.5.37 + NSP9_T21I	7 (78) *	15/12/2022	23/04/2023	129
XBB.2.3.13 + NSP3_Q167R	6 (60) *	28/02/2023	31/07/2023	153
XBB.2.3.13 + NSP2_A357S + NSP6_F235L + S_A701V	13 (93) *	12/03/2023	24/04/2023	43
XBB.1.5.71 + NSP15_P205S + NSP2_L24F	5 (83) *	10/05/2023	27/09/2023	140
BA.4.1 + NSP13_V157L	7 (88) *	09/06/2022	15/08/2022	67
BE.1 + ORF3a_T270I + ORF3a_V273L	6 (86) *	10/07/2022	08/09/2022	60
BF.7 + NSP4_A128V + S_Q675H + S_P1263Q	10 (19)	29/06/2022	01/12/2022	155
CK.2.1.1 + S_V1264L + ORF8_D119Y	5 (63)	03/10/2022	21/12/2022	79
EL.1 + NSP2_T103I + NSP3_T720I	35 (90)	16/11/2022	19/05/2023	184
EF.1.2 + S_D253G + NSP4_V13I + ORF3a_D155Y	10 (67)	20/12/2022	16/03/2023	86
XBB.2.3.13 + NSP2_A357S	12 (16)	05/02/2023	09/06/2023	124
EL.1 + NSP14_A320T + NSP2_G265S + NSP3_T970M	6 (75)	14/02/2023	19/05/2023	94
DV.7.1 + NSP14_N71S	8 (57)	28/04/2023	02/09/2023	127
XBB.1.5 + NSP3_V1673I + ORF3a_L140F	13 (81)	08/06/2023	19/09/2023	103
EG.6.1 + NSP3_S1428L	11 (79)	16/06/2023	02/10/2023	108
DV.7.1 + NSP8_P10S + NSP3_D1764G	5 (16)	23/06/2023	10/09/2023	79
EG.5.1.5 + S_N354K + N_P151S	12 (46)	13/07/2023	26/08/2023	44
HV.1 + NSP14_P203L	6 (29)	05/08/2023	13/10/2023	69
JN.1 + S_F486V	6 (75)	29/10/2023	25/01/2024	88

** detected only in Asturias; * detected also in Spain.

**Table 6 viruses-17-01531-t006:** Mutations fixed in Asturias used to define the genetic characteristics of SARS-CoV-2 variants of interest by the WHO.

	Variant	Alpha	Beta	Gamma	Delta	Zeta	Eta	Theta	Iota	Kappa	Lambda	Mu	Epsilon	Omicron
Mutation														
E_T9I				X				X			X			X
M_A63T														X
N_G204R		X		X		X		X			X			X
N_P13L											X			X
N_R203K		X		X		X		X			X			X
ORF1a_T3255I											X	X		X
RdRp_P323L		X	X	X	X	X	X	X	X	X	X	X	X	X
S_D614G		X	X	X	X	X	X	X	X	X	X	X	X	X
S_E484K			X	X		X	X	X				X		
S_H655Y				X										X
S_K417N			X											
S_N501Y			X	X				X				X		
S_N679K														X
S_P681R					X					X				
S_Q954H														X
S_S373P														X
S_T478K					X									X

## Data Availability

Data availability in GISAID (https:www.gisaid.org/).

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
