# Peer review of "New Circulating Variants of SARS-CoV-2 in Asturias During the Period (2022–2024)"

_viruses, 2025, doi:10.3390/v17121531_

Round 1
Reviewer 1 Report
Comments and Suggestions for Authors
The authors sequence and analyze 4,001 SARS-CoV-2 genomes from Asturias (2020–July 2024), focusing on mutation counts, “fixed” changes, lineage diversity, and “new circulating variants” (NCVs). The study’s strengths include its longitudinal regional coverage (with a post-Omicron emphasis) and a comprehensive catalog of variants/mutations across genes that links several fixed mutations to WHO VOIs/VOCs—providing a practical reference for surveillance and comparative analyses.
The manuscript remains largely descriptive. Counting mutations, lineages, and NCVs is useful, but the claims should be tested quantitatively.
- The rule that defines a mutation as “fixed” when it reaches 95% of samples in one month should be tested with different cut-off points (for example, 85%, 90%, and 95%) to see if the results still hold. Likewise, the rule calling a mutation “major” when it appears in more than 5% of samples should consider how many samples were tested each month, since small sample sizes can make the percentage misleading.
- Explain how each new circulating variant (NCV) was defined in relation to the official Pango lineage system.
- The manuscript requires substantial language polishing and consistency editing. There are numerous typographical and formatting issues—such as incorrect capitalization (“Sars-Cov2” instead of SARS-CoV-2) and broken URLs (“h ps://”). Full language edit is suggested. Especially in Methods and Discussion.
- There are two Table 1 for different content. Ensure consistent labels and captions.
- Use “non-synonymous amino-acid changes” (not “aminoacid”), and standard gene names (“RdRp” instead of “RDPD”).
Author Response
|
1. Summary |
|
|
Thank you very much for taking the time to review this manuscript. Please find the responses below and the corresponding revisions/corrections highlighted/in track changes in the re-submitted files.
|
|
|
2. Point-by-point response to Comments and Suggestions for Authors |
|
|
Comments 1: The rule that defines a mutation as “fixed” when it reaches 95% of samples in one month should be tested with different cut-off points (for example, 85%, 90%, and 95%) to see if the results still hold. Likewise, the rule calling a mutation “major” when it appears in more than 5% of samples should consider how many samples were tested each month, since small sample sizes can make the percentage misleading.
|
|
| Response 1: We agree with this comment. The definition of cut-off points is often debatable. Strictly speaking, a mutation is considered fixed in the population when it is present in 100% of samples. However, due to the possibility of sequencing errors, we have decided to be somewhat less strict. We believe that testing other cut-off points in this case would only lead us to analyze mutations that may have become fixed in subpopulations and overestimate the time it takes for them to become fixed. Prevalent mutations refer to the entire sequenced population, as we are looking for mutations that may be relevant to the population as a whole, not just at a specific stage or in a specific subpopulation. The higher the proportion of a mutation in the population, the more likely it is to be important. However, given the low number of mutations, we decided to use a very low cutoff point, which will likely include mutations of little importance.
|
|
|
Comments 2: Explain how each new circulating variant (NCV) was defined in relation to the official Pango lineage system.
|
|
|
Response 2: The Pango lineage system is based on the study of SNPs and monophyletic clades with more than five sequences. The sentence on page 3 in Materials and Methods is completed. Possible new lineages were defined for study based on SNPs and monophyletic clades with more than five sequences (According to the proposed dynamic nomenclature for SARS-CoV-2 lineages at https://github.com/cov-lineages/pango-designation).
Comments 3: The manuscript requires substantial language polishing and consistency editing. There are numerous typographical and formatting issues—such as incorrect capitalization (“Sars-Cov2” instead of SARS-CoV-2) and broken URLs (“h ps://”). Full language edit is suggested. Especially in Methods and Discussion.
Response 3: Incorrect capitalization “Sars-Cov2” instead of SARS-CoV-2 corrected. and broken URLs (“h ps://”). Broken URLs (‘h ps://’) not found in the manuscript.
Comments 4: There are two Table 1 for different content. Ensure consistent labels and captions.
Response 4: All numbers in the tables have been modified so that they follow a numerical order throughout the manuscript.
Comments 5: Use “non-synonymous amino-acid changes” (not “aminoacid”), and standard gene names (“RdRp” instead of “RDPD”).
Response 5: ‘aminoacid’ changed to’ amino-acid changes’. ‘RdRp’ instead of ‘RDPD’ changed.
|
|
Reviewer 2 Report
Comments and Suggestions for Authors
This manuscript presents a comprehensive genomic surveillance study of SARS-CoV-2 in Asturias, northern Spain, covering the period 2022–2024. By analyzing over 4,000 viral genomes and identifying novel mutations and lineages, the authors contribute valuable regional data to the global effort of monitoring SARS-CoV-2 evolution.
The study is well-designed and addresses a relevant public health topic. However, the manuscript would benefit from improved clarity in several sections, a more concise presentation of results, and a deeper contextualization of the findings within the broader framework of global genomic surveillance. Extensive language editing is also required to enhance readability and scientific precision.
Major Comments
- The abstract should be more concise, emphasizing the key quantitative results (e.g., number of novel circulating variants, mutation fixation time) and main conclusions.
- While the study adds valuable regional data, the authors should more clearly highlight the biological or epidemiological insights that extend beyond previous genomic surveillance reports. Please also reference other regional or national surveillance efforts worldwide to better position this work within the global context. These points could be incorporated at the end of the Introduction and revisited in the Discussion/Conclusion.
- Specify the total number of collected samples versus those successfully sequenced. Indicate whether Ct value or viral load influenced inclusion criteria.
- The sequencing and bioinformatic pipelines are well described, but the rationale for key thresholds (e.g., “mutations >5% of the viral population,” fixation at >95%) should be better justified, ideally supported by citations.
- Clarify the timeframes defining the two study periods (e.g., 2020–2022 vs. 2022–2024) and explain how these correspond to the “pandemic” and “post-pandemic” phases.
- Indicate whether any of the newly detected circulating variants (NCVs) exhibited signatures suggestive of adaptive or transmission-related changes compared to globally circulating strains.
- The discussion should be more concise and clearly centered on the study’s own results rather than broad background information. It would also be valuable to relate the findings to selective pressures (e.g., immune evasion, vaccination coverage) and to compare lineage turnover rates with other European regions. Several statements currently lack references—please ensure appropriate citation support.
- Figures:
- Figure 1: Adjust the y-axis scale to improve visualization of the bars and enhance image resolution.
- Figure 2: Consider splitting the figure into two panels to clearly separate the two study periods.
- Figure 3: Clarify the color scheme in the legend for panels A and B.
- Figure captions: Expand captions to make the figures understandable without referring to the main text.
- Strengthen the conclusion by discussing the implications of continued local variant emergence for national and European surveillance strategies.
Minor Comments
- Lines 22–23: Text highlighted in grey, please remove.
- Line 36: Replace SARS-Cov-2 with SARS-CoV-2 (apply consistently throughout).
- Line 38: Replace COVID with COVID-19 (apply consistently throughout).
- Line 76: “RDPD” should be RdRp (RNA-dependent RNA polymerase); correct throughout the manuscript.
- Line 80: Remove the extra period.
- Line 130: Modify to “fixed mutation.”
- Table 4: Correct NVC to NCV.
- Lines 276–280: Text highlighted in grey, please remove.
Recommendation
Major Revision.
The manuscript provides relevant and robust genomic surveillance data, but revisions are required to improve clarity, language, and integration with existing literature. With these improvements, the study would represent a valuable contribution to understanding SARS-CoV-2 evolution in the post-pandemic phase.
Comments on the Quality of English Language
Language editing is required to enhance readability and scientific precision
Author Response
Response to Reviewer 2 Comments
|
1. Summary |
|
|
|
Thank you very much for taking the time to review this manuscript. Please find the responses below and the corresponding revisions/corrections highlighted/in track changes in the re-submitted files.
|
||
|
2. Point-by-point response to Comments and Suggestions for Authors Major Comments
|
||
Comments 1: The abstract should be more concise, emphasizing the key quantitative results (e.g., number of novel circulating variants, mutation fixation time) and main conclusions.
Response 1: new abstract
The ability of a virus to adapt is key to its survival, and this is achieved through mutation, which allows the virus to change and adapt to new environments. To capture the full extent of SARS-CoV-2 diversity in Asturias, samples obtained from nasopharyngeal swabs were characterized using whole genome sequencing. Between 2020 and July 2024, a total of 4,001 sequences were analysed and 5,302 mutations were identified. An increase in the positivity rate was observed between 2022 and 2024 in children under 1 year of age. During this period, 55 new circulating variants belonging to 41 pangolin lineages were detected: 24 originated throughout the world and 31 in Asturias (10 were only detected in the region, eight also in Spain and 13 around the world). Thirty-one new non-synonymous mutations were fixed in the viral population 250 ± 46 (93–620) days after their appearance. During seasonal SARS-CoV-2 circulation, surveillance systems developed during the pandemic continue to detect new indigenous and imported variants without indicating an increase in severity
Comments 2: While the study adds valuable regional data, the authors should more clearly highlight the biological or epidemiological insights that extend beyond previous genomic surveillance reports. Please also reference other regional or national surveillance efforts worldwide to better position this work within the global context. These points could be incorporated at the end of the Introduction and revisited in the Discussion/Conclusion.
Response 2: We would be grateful if you could point us towards other surveillance initiatives similar to ours. The studies we found are based on the monitoring of new variants that have been accepted by the Pango lineage system. Our approach, on the other hand, is based on a systematic search for new variants.
Comments 3: Specify the total number of collected samples versus those successfully sequenced. Indicate whether Ct value or viral load influenced inclusion criteria.
Response 3: Added to results page 3
A total of 4,001 sequences were analysed (randomly selected from 533,769 positive SARS-CoV-2 samples with ct<27) in Asturias (northern Spain) between 2020 and July 2024.
Comments 4: The sequencing and bioinformatic pipelines are well described, but the rationale for key thresholds (e.g., “mutations >5% of the viral population,” fixation at >95%) should be better justified, ideally supported by citations.
. Response 4: The definition of cut-off points is often debatable. We did not find any citations to choose a specific cutoff point. Strictly speaking, a mutation is considered fixed in the population when it is present in 100% of samples. However, due to the possibility of sequencing errors, we have decided to be somewhat less strict. We believe that testing other cut-off points in this case would only lead us to analyze mutations that may have become fixed in subpopulations and overestimate the time it takes for them to become fixed. Prevalent mutations refer to the entire sequenced population, as we are looking for mutations that may be relevant to the population as a whole, not just at a specific stage or in a specific subpopulation. The higher the proportion of a mutation in the population, the more likely it is to be important. However, given the low number of mutations, we decided to use a very low cutoff point, which will likely include mutations of little importance.
Comments 5: Clarify the timeframes defining the two study periods (e.g., 2020–2022 vs. 2022–2024) and explain how these correspond to the “pandemic” and “post-pandemic” phases.
Response 5: Added to introduction page 2
In June 2022, the Spanish Ministry of Health proposed infection control and surveillance measures that reduced pressure on cases of mild or asymptomatic illness and their contacts[5]. Consequently, all respiratory viral illnesses, including those caused by SARS-CoV-2, must now be considered as posing the same risk.
Comments 6: Indicate whether any of the newly detected circulating variants (NCVs) exhibited signatures suggestive of adaptive or transmission-related changes compared to globally circulating strains.
Response 6: changed in discussion page 12
Fortunately, all newly detected circulating variants appear to be extinct and not exhibited signatures suggestive of adaptive or transmission-related changes
Comments 7: The discussion should be more concise and clearly centered on the study’s own results rather than broad background information. It would also be valuable to relate the findings to selective pressures (e.g., immune evasion, vaccination coverage) and to compare lineage turnover rates with other European regions. Several statements currently lack references—please ensure appropriate citation support.
Response 7: We considered removing the description of mutations that are fixed or generate new variants in Asturias, and finally decided that the discussion was not too long, although it can be removed if necessary.
It would be really interesting to relate the findings to selective pressures and compare lineage turnover rates with those of other regions. However, given that the new lineages detected in Asturias do not appear to show signs of adaptation or be related to transmission, we will leave this comparison for future studies.
Please indicate any statements that lack references
- Figures:
- Figure 1: Adjust the y-axis scale to improve visualization of the bars and enhance image resolution.
Changed by
- Figure 2: Consider splitting the figure into two panels to clearly separate the two study periods.
Mutations that appear in the first period are fixed in the second, so two panels were unbalanced and we thought it looked better that way.
- Figure 3: Clarify the color scheme in the legend for panels A and B.
There are 313 colors in the legend of panel A, so we did not include it; it is only a representation of the existing diversity. In panel B, only the major lineages are indicated.
Added ‘only the major lineages are indicated’ to the panel B legend
- Figure captions: Expand captions to make the figures understandable without referring to the main text.
Figure1. Rate of main mutations circulating in Asturias that have been fixed ( ) and lost ( ) in the population in each gene during Covid (March 2020 to June 2022) and post-Covid period (July 2022 to July 2024).
Figure 2. Number of major SARS-CoV-2 mutations (>5% of the viral population) that became fixed in Asturias during COVID (March 2020 to June 2022) and the post-COVID period (July 2022 to July 2024). The figure shows the dates of first detection and fixation
Figure 3. Number of pangolin lineages (313 color scheme not included) detected in Asturias in the post-COVID period (July 2022 to July 2024) (A). Percentage of pangolin lineages mostly detected over time (>1%) in the post-COVID period (July 2022 to July 2024), only the major lineages are indicated (B).Number of new circulating variants originated in Asturias or imported and incidence of SARS-CoV-2 in the post-COVID period (July 2022 to July 2024)(C).
Figure 4: Dated phylogeny of the new circulating variant, XBB.1.5.8+NSP14_V328F, which was imported and belongs to the same transmission clade in the region. The days until the date of the most recent sequence were used as the sampling date.
- Strengthen the conclusion by discussing the implications of continued local variant emergence for national and European surveillance strategies.
Added to conclusion
The discovery of new local variants that can be imported and exported emphasises the importance of local surveillance in informing public health responses relating to diagnostics, vaccination strategies and health policies at local, regional and global levels.
Minor Comments
- Lines 22–23: Text highlighted in grey, please remove.
Done
- Line 36: Replace SARS-Cov-2 with SARS-CoV-2 (apply consistently throughout).
Done
- Line 38: Replace COVID with COVID-19 (apply consistently throughout).
Done
- Line 76: “RDPD” should be RdRp (RNA-dependent RNA polymerase); correct throughout the manuscript.
Done
- Line 80: Remove the extra period.
Done
- Table 4: Correct NVC to NCV.
Done
- Lines 276–280: Text highlighted in grey, please remove.
Done
Round 2
Reviewer 1 Report
Comments and Suggestions for Authors
The authors have addressed all the comments. I have no further comments.